# Application of Emerging Techniques in Reduction of the Sugar Content of Fruit Juice: Current Challenges and Future Perspectives

**DOI:** 10.3390/foods12061181

**Published:** 2023-03-10

**Authors:** Magdalena Cywińska-Antonik, Zhe Chen, Barbara Groele, Krystian Marszałek

**Affiliations:** 1Wacław Dąbrowski Institute of Agricultural and Food Biotechnology, Department of Fruit and Vegetable Product Technology, 02532 Warsaw, Poland; magdalena.cywinska@ibprs.pl (M.C.-A.); zhe.chen@ibprs.pl (Z.C.); 2Polish Association of Juice Producers, 02532 Warsaw, Poland; b.groele@kups.org.pl; 3Department of Food Technology and Human Nutrition, Institute of Food Technology and Nutrition, College of Natural Science, University of Rzeszow, Zelwerowicza 2D, 35601 Rzeszow, Poland

**Keywords:** low sugar fruit juice, calorie-reduced juice, nanofiltration, enzymatic reaction

## Abstract

In light of the growing interest in products with reduced sugar content, there is a need to consider reducing the natural sugar concentration in juices while preserving the initial concentration of nutritional compounds. This paper reviewed the current state of knowledge related to mixing juices, membrane processes, and enzymatic processes in producing fruit juices with reduced concentrations of sugars. The limitations and challenges of these methods are also reviewed, including the losses of nutritional ingredients in membrane processes and the emergence of side products in enzymatic processes. As the existing methods have limitations, the review also identifies areas that require further improvements and technological innovations.

## 1. Introduction

The typical diet abounds with high sugar in food products, including sweets, sweetened beverages, and sweetened dairy products. During the last fifty years, sugar consumption in the world has tripled. According to world data, people consume more than 500 calories daily from added sugar [1]. There is a correlation between the consumption of sugar-rich products and the risk of diet-dependent diseases, including diabetes type 2, cardiovascular diseases, obesity, insulin resistance, and certain types of cancer [2]. In 2020, the World Health Organization (WHO) announced that heart diseases remain the world’s leading cause of death, accounting for even 16% of all deaths globally [3], demonstrating that improving human nutrition is one of the most important roles in preventing diseases. The public’s growing interest in well-being and health improvement increased the interest in the replacement of table sugar (refined sugar) with sugars from natural sources, such as grapes, honey, date fruit syrup [1], or sugarcane juice [4], providing many health benefits. Consuming fruit and vegetables prevents many of civilization’s diseases, and promoting fruit and vegetable consumption, or “5 a day” as a healthy diet, is important. The recommendation of consuming five servings of fruits and vegetables daily (including a maximum of 150 mL of juice as one portion) is well known [5].

Fruit juices are a good source of vitamins (especially vitamin C) and antioxidant compounds—polyphenols, carotenoids, or folate [6,7]. However, juices also contain natural sugars such as glucose, fructose, and sucrose. In many cases, the sugar content in natural juice (no added sugar) may be close to or even higher than that in popular non-fruit drinks such as sugar-sweetened beverages (SSBs) [8,9,10]. However, contrary to added sugars, sugars naturally present in whole foods contribute to preserving health and a well-balanced diet [1]. Even though the sugar levels in fruit juices and SSBs are insignificant, the influence on human health is very significant due to other nutritional compounds in fruit juices, such as dietary fiber, polyphenols, vitamins, and minerals. Some of these compounds, such as fiber, can slow down the bio-accessibility of sugars into the circulatory system by slowly releasing them through the digestive process [11,12].

In the EU, producers of fruit juices are obliged to meet the requirements of the techniques used for producing juices, adding food additives, and the preservation of the quality of the final product. The primary document obligatory for all of the juice producers in the EU is the Juice Directive—Council Directive 2001/112/EC (the amendment 2012/12/EU from 19 April 2012 with the last changes currently in force). According to the requirements contained in this document, fruit juice can be produced only from fruits that are sound, ripe, and fresh or preserved by chilling or freezing. Only physical methods, i.e., mechanical extraction, are allowed during production. According to the Juice Directive, it is possible to add substances during production, such as edible gelatin, tannins, charcoal, silica sol, bentonite as an adsorbent clay, and pectolytic, proteolytic, and amylolytic enzymes, for technological reasons. Fruit juice is a fermentable but unfermented product obtained from the edible part of at least one type of fruit, having the characteristic color, flavor, and taste of the fruit from which it comes [13]. The physicochemical quality of fruit juices should be within the limits set by the Code of Practice of the European Fruit Juice Association—AIJN [14], such as the appropriate minimum total soluble solids, sugars concentration, acidity, vitamin C, formol number, and others. The AIJN Code is a voluntary code of practice used by the EU’s processors and traders of fruit juices.

Most of the total sugar content in fruit juices are mono sugars such as glucose or fructose. It must be highlighted that according to EU regulations, it is forbidden to add sugars to fruit juice production [13]. The sugars, together with other bioactive compounds, can only come directly from the fruits; despite this, the WHO recommends reducing the intake of free sugars to less than 10% of the total energy intake [15]. Consumers are looking for food products with health-promoting impacts on their bodies, such as fruit juices and drinks. Therefore, there is an increasing market for functional beverages with additional health-promoting properties [16]. The research described by Olewnik-Mikołajewska (2016) shows that consumers prefer products with fewer ingredients that have an adverse effect on the body [17]. This contributes to the global trends for promoting products with the best possible nutritional values and functional foods with nutrition claims such as being “energy-reduced” or “sugar-reduced”. According to current legal regulations, a claim relating to reducing the energy or sugar content may only be made when it is by at least 30% compared with similar fruit juices [18]. Reducing caloric intake from fruit juices is a future challenge for scientists considering that changes in other bio-functional properties and natural composition of other nutrients should be insignificant [19]. The production of juices with reduced sugar content might be very interesting from a nutritional point of view and fits very well with WHO recommendations [15].

It should be emphasized that juices are not a main source of sugar in the human diet. The main contributors to excessive sugar consumption are sweets, cookies, and carbonated and non-carbonated beverages [20,21,22]. Depending on the country concerned, consuming sugar-added products is sometimes 50% of the energy intake for children and 60% for adults. For beverages, it equals about 30% of the energy intake for adults and children [22]. Consumption of added sugars worldwide is the greatest among children and decreases with age [23].

The EU’s juice consumption is about 12 L per capita per year (2017). In Poland, on average, it is about 15 L of juices per capita per year (2017) [24]. Since juice contains an average of 10% of total natural sugars, only 1.5 kg of sugar per year can be transferred to the human body through juices. Moreover, Poland is in the top 10 of countries with the highest total sugar intake, which equals about 45 kg per capita per year (2017) [25], demonstrating that excessive sugar consumption is directly related to incorrect nutritional habits, mainly associated with not maintaining a healthy well-balanced diet, not juice consumption. Nevertheless, scientists are looking for different ways to improve the health properties of juices. Hence, this review presents simple methods to produce juice with reduced sugar content in fruit juices. Therefore, we will summarize the current knowledge and new technological aspects of low-caloric juices (LCJ).

## 2. Techniques Used in Reducing the Sugar Content of Juices

### 2.1. Conventional Techniques

One of the easiest techniques is mixing two different juices in appropriate proportions. Two juices with significant differences in °Brix values may give the final product a 30% sugar reduction. For example, the combination of apple juice (with a minimal °Brix for direct juice equal to 10.0) and coconut water (with a °Brix value for direct juice equal to 3.6) in a 1:1 volume ratio would reduce the concentration of sugars by a minimum of 30% (Table 1) [14].

Mixing juices benefits the beverages’ functional properties, nutritional value, and sensory properties [26]. Utilization of some highly nutritive fruits (such as acerola, known for its high content of vitamin C [27] or cranberry, known as a rich source of phenolic compounds [28]) with a low °Brix value equal to 6.0 [14] is very limited due to their bitterness, high acidity, or astringency [29]. Mixing them with high-sugar fruit juices results in an increase in their utilization and the obtaining of juices with reduced sugar content enriched with high nutrients.

Moreover, in the case of fruit with a high viscosity of pulp and high °Brix value, such as with bananas [30], adding another juice with a much lower °Brix value leads to a significant reduction in the concentration of sugars even by half. It reduces viscosity without the need for the addition of water. Table 1 compares sugar reduction levels after mixing in an appropriate volume ratio of two juices with significantly different °Brix values. The main disadvantage of this method is the inability to obtain juice consisting of only one type of fruit or juice from fruits with similar °Brix values. Moreover, this technique only enables the reduction of the °Brix of the mixed juice compared to single-component fruit juice with higher °Brix.

### 2.2. Membrane Filtration Processes

Among the membrane technologies that have been used to separate bioactive compounds such as sugars from agri-food industry products are microfiltration (MF), ultrafiltration (UF), and nanofiltration (NF). However, it is observed that juices show complex composition, contributing to a significantly greater decrease in permeate flux than in solutions of simple composition, such as fructose solutions [31]. To improve the efficiency of separating low molecular weight molecules from valuable higher molecular weight molecules and an increase in permeate flux, membrane processes can be used in the diafiltration (DF) mode. It enables more molecules to contact the membrane and pass through its pores [32]. In membrane techniques, there are two main filtration modes: dead-end filtration and cross-flow filtration (Figure 1) [33]. During dead-end filtration, the feed stream moves perpendicularly to the membrane, but the fouling phenomenon restricts the filtration rate or flux (Figure 1a). In cross-flow filtration, the solution flows tangentially over the membrane surface (Figure 1b).

Various compounds such as sugars, proteins, and colloidal materials are soluble in fruit juices. The concentration of these compounds is measured using a parameter known as the total soluble solid content (TSS) [34,35]. The separation of biomolecules in heterogeneous suspensions by membrane techniques is based on certain physicochemical properties, such as molecular weight, charge, monomer composition, chemical affinity, or dielectric properties of soluble solids [36,37]. Membrane processes offer high separation selectivity, quick kinetics of the reaction, and minimal energy consumption [38]. The main disadvantage of separation techniques, which can reduce the potential of this technology for application in the industry, is the susceptibility to the accumulation of plant tissue and fiber on the membrane surface. It may reduce membrane lifetime and deteriorate its separation capacity [32]. In the membrane process, high molecular weight molecules such as pectins, starches, proteins, phenolic compounds, polysaccharides, and colloidals were retained on the membrane surface to form a gel-type foulant layer, thus reducing membrane pore size and growing with filtration time [39,40,41]. Since most of these compounds could be separated by the membranes during processing, the nutritional quality of the juice obtained in permeate after processing is inadequate, especially when all phenolic compounds and fibers are separated. Moreover, only clear juices can be obtained in the permeate due to filtration processes. However, compared to conventional techniques, such as clarification or thermal concentration, membrane processes ensure better control over the loss of nutritive compounds present in fruit juice [42].

#### 2.2.1. Microfiltration and Ultrafiltration

In the food industry, microfiltration and ultrafiltration are applied to clarify juices [43]. Although it is difficult to separate sugar in juices during MF and UF processes due to the high membrane pore size, the pore size of the membrane was affected by the feed juices. Reducing membrane pore size by forming a gel-type layer “secondary membrane” [41,44] can keep sugar molecules on the membrane surface, promoting the separation of the sugar in juices. The pore size in the microfiltration process is sufficient to retain sucrose by membrane and foulant layer partially. In contrast, solutes with less molecular weight than sucrose are freely permeable through the membrane [41]. Therefore, MF and UF were also selected to reduce the sugar content in juices (Table 2). Using membranes with smaller pore sizes generally contributes to the increasing reduction of smaller soluble solids. This dependence was shown in the study of Ghosh et al. (2018), where the reduction of membrane pore size in the case of jamun juice resulted in decreasing TSS content in the permeate [45]. However, in the case of the filtration of pomegranate juice, microfiltration and ultrafiltration processes had a comparable effect on TSS content [46]. However, the combination of these processes (UF process applied to MF-treated juice) was more effective in decreasing TSS content (Table 2) [42].

To maintain the permeate flux and improve the efficiency of the filtration process, a diafiltration (DF) combined with UF was performed through several investigations. This technique promotes the dissolution of macromolecules deposited on the membrane surface, thereby declining the membrane fouling and improving the permeate flux of the membrane [58]. The diafiltration step could be worthy of attention for reducing the sugar content in juices and producing healthier concentrates [59]. Samborska et al. (2018) examined the impact of UF and DF on reducing the sugar content in cloudy apple-cranberry juice. The authors treated retentate as a final product of ultrafiltration that enables obtaining cloudy juice. As a result of the additional DF step, the concentration of sugars in the retentate was significantly higher [53]. Generally, despite all of the aforementioned advantages, the diafiltration technique cannot be used in fruit juice technology production due to the addition of water to the product. According to the Juice Directive, adding water to fruit juice is prohibited in the EU [13].

#### 2.2.2. Nanofiltration

Nanofiltration (NF) offers higher retention for compounds with lower-molecular weight, such as sugars, more so than UF or MF [60]. As a result of the process (especially combined with DF), it is possible to remove sugars almost completely (Table 2). However, Pruksasri et al. (2020) emphasize that to ensure a satisfactory degree of preservation of bioactive compounds, sugar reduction should be at most 30% [19]. Despite being capable of retaining compounds with a low molecular weight, using the nanofiltration process is extremely challenging when separating molecules with similar properties, such as those with equal molecular weights. Morthensen et al. (2015) used an approach to confront this challenge. The authors separated xylose from a mixture of xylose and glucose—two mono sugars with a similar structure, charge, and size. For this purpose, they used an enzymatic process (glucose oxidase and catalase enzymes) to convert glucose into gluconic acid. The NF270 polyamide flat sheet membrane with MWCO of 150–200 Da was applied during the study. The separation of these compounds was based on sieving effects and on negative charge repulsions between the surface of the membrane and gluconate (the conjugate base of gluconic acid). The feed molar ratio of xylose to gluconic acid was 9:1. Under these conditions, after the process, the xylose separation factor increased from 1.4 (obtained in the xylose/glucose system) to 34 (obtained in the xylose/gluconic acid system). Glucose retention was 68%, whereas gluconic acid was almost fully retained under the same process conditions [61,62]. The separation factor is defined as the ratio of the concentration ratio of two substances in the permeate and retentate.

Nanofiltration is used either as a major process or in combination with other separation processes [63]. A combination of several methods can improve the efficiency of the process (Table 2). Reduction of sugar concentration can be obtained through a combination of ultrafiltration and nanofiltration membranes. In the first stage, fruit juice is filtered by an ultrafiltration membrane to remove suspended solids. As a result of the ultrafiltration stage, a clarified juice can be obtained in the permeate. It reduces the formation of the fouling layer. In the next stage, UF permeate is subjected to the NF process by a selective membrane impermeable to sugars to produce a poor sugar permeate. Finally, the obtained solution can be mixed with UF retentate and/or a portion of the untreated juice (if desired) to produce a juice with a reduced concentration of sugars [64].

### 2.3. Enzymatic Process

Specific enzymes can indicate and/or accelerate chemical reactions leading to a reduction in the sugar content in fruit juices by biological catalysts. The use of particular enzymes reduces the concentration of selected sugars in juices. Glucose can be converted into gluconic acid due to the use of two different enzymes: glucose oxidase and catalase [65]. Fructose can be converted into D-allulose (one of the rare sugars and a low-caloric sugar) by using D-allulose 3-epimerase [66] or into sorbitol by using a glucose-fructose oxidoreductase [67,68]. Moreover, the application of glucosyltransferase enables the conversion of sucrose, in the presence of glucose, into gluco-oligosaccharides [69], whereas the application of fructosyltransferase leads to the formation of fructo-oligosaccharides from sucrose [70].

Applying specific enzymes can improve the efficiency and quality of food products with low costs by improving nutritional safety, reducing energy costs related to the process, and leading to process optimization [71]. Biotechnology processes catalyzed by enzymes are carried out under smooth pH and relatively low temperature, which is especially important in the case of sensitive substances, and they exhibit high stereochemical selectivity [65]. On the other hand, some of the enzymatic processes require preliminary steps: liquefaction, gelatinization, saccharification, and purification or hydrolysis of polysaccharides. Hydrolysis reaction produces undesirable monosaccharides and/or oligosaccharides in the final product [72]. Moreover, free enzymes have some limitations—the most important is that their recovery and reuse are difficult [73]. Biocatalytic enzymes can be immobilized, which enables, among other things, a good recovery and reuse of the enzyme and improves the enzyme’s stability [74] and activity under different reaction conditions [75,76]. On the other hand, due to the permanent connection of the enzyme with the carrier in the immobilization process, removing the deactivated enzyme is difficult, which may result in the inability to reuse the matrix [77]. The most important factor that affects the effectiveness of the preparations after their immobilization is the type of enzyme immobilization (Figure 2) [78] and the carrier used for immobilization (Table 3) [78,79,80,81]. The immobilized enzyme can be removed from the reaction solution by centrifugation or filtration [82]. However, among the main problems associated with the immobilization of enzymes, loss of enzyme activity, especially in the presence of macromolecular substrates, should be mentioned. This phenomenon is caused by the substrate’s limited availability in the enzyme’s active site. As a result, enzyme activity can be limited to accessible surface groups of the substrate. One of the useful approaches for solving this problem is using hydrophilic and inert spacer arms [83].

In 2021, the Israeli start-up Better Juice and the GEA Process Engineering Group published their method for reducing the sugar content of any fruit or vegetable juices that contain sugar, including apple, strawberry, or orange juices. This method uses immobilized microorganisms containing non-genetically modified enzymes to convert the juices’ sugars (glucose, fructose, and sucrose) into dietary fiber, indigestible polysaccharides, or other compounds. The authors used, among other things, immobilized and dead *Zymomonas mobilis* or cells to reduce the concentration of glucose and fructose and the yeast *Aureobasidium pullulans* to reduce sucrose concentration. The activity of enzymes within the dead microorganisms has to be preserved. Fructose, glucose, and sucrose were converted into sorbitol, gluconic acid, and dietary fiber. The results demonstrated that texture, smell, and vitamin composition remained unchanged compared with the unchanged juice. The taste of the juice keeps its palatability, only with reduced sweetness. Using live or dead microorganisms does not have a negative effect on taste. The sugar content of the juice can be significantly reduced compared with the starting food product containing sugar. Sucrose, glucose, and fructose content can be reduced by at least 5% to more than 95%. Among the disadvantages of this process is a negligible ethanol concentration in a food product (less than 0.5% *v*/*v*). Ethanol can be removed from the food product using any known method [84,85].

#### 2.3.1. Gluconic Acid

More than one enzyme is required to achieve the proper functioning of many catalytic transformations, which has attracted increasing interest in applying multi-enzymatic transformation technologies. Complex, multi-enzymatic systems, especially immobilized ones, can carry out complex catalytic changes. Multi-enzymes can be immobilized on separate carriers or one collective carrier to participate in this complex catalytic transformation [77]. The advantage of multi-enzymes systems is that they reduce the demand for cost, time, and the chemicals used for product recovery. Moreover, it can maintain a minimum concentration of harmful or unstable compounds [65,77,86].

One of the multi-enzymatic systems is invertase, catalase, and glucose oxidase immobilized on a porous membrane. It can convert sucrose into gluconic acid and fructose [65] in fruit juices. Invertase, mainly from *Saccharomyces cerevisiae* (Baker’s yeast), can produce inverted sugar—an equal mixture of fructose and glucose [87,88]. Glucose oxidase oxidizes glucose into hydrogen peroxide and gluconic acid [65]. Moreover, glucose oxidase can improve the organoleptic properties of food products, such as improving the color stability of grape juice or enhancing aroma and taste (non-flavonoid phenolic content). Glucose oxidase exhibited a beneficial effect in modulating non-enzymatic browning during the processing of food [89]. Catalase (one of the most common enzymes [90]) is used as an auxiliary enzyme in the conversion reaction of glucose into gluconic acid [65]. Catalase is used to degrade the toxic hydrogen peroxide produced during the glucose oxidase conversion reaction. Hydrogen peroxide degradation minimizes toxic compounds’ presence and eliminates the inhibitory effect of H_2_O_2_ on the glucose oxidase conversion reaction [86].

The sequence of the multi-enzyme conversion can be illustrated through the equations, where S—sucrose, F—fructose, G—glucose, and GA—gluconic acid:
(1)$$S\;\overset{\left(\mathrm{invertase}\right)}{\to }\;F+G$$
(2)$$G\;\overset{\left(\mathrm{glucose}\;\mathrm{oxidase}\right)}{\to }\;G+\mathrm{GA}+H_{2}O_{2}$$
(3)$$2{\;H}_{2}O_{2}\overset{\left(\mathrm{catalase}\right)}{\to }2{\;H}_{2}O+{\;O}_{2}$$

Glucose oxidase has been proposed to optimize the glucose content of grape juice before fermentation to produce low-alcohol wines. As a result of the reaction, an 87% reduction of glucose content and 73 g/L of gluconic acid have been obtained [89].

In the study conducted by Laue et al. (2019), the multi-enzymatic conversion of sucrose into gluconic acid and fructose in apple juice was carried out. The authors added invertase to juice at an ambient temperature, which achieved the sucrose concentration after conversion of <0.01 gL/L. Subsequently, the addition of glucose oxidase/catalase with the simultaneous addition of oxygen (supply of the oxygen into the reaction tank was constant—3 mg/L) was noted. The pH of the juice was maintained at 3.6–4.6 by adding calcium hydroxide and potassium hydroxide. In order to terminate the activity of the enzymes, the supply of oxygen was stopped. In the last stage, irreversible denaturation of all enzymes was conducted by boiling. Further addition of potassium/calcium hydroxide enabled the optimization of the organoleptic properties. The pH of the final mixture was 4. Control and verum were similar in appearance, texture, flavor, and color [91].

The study described by Laue et al. (2019) also investigated the impact of converting free glucose and glucose from sucrose to gluconate/D-gluconolactone in apple juice on postprandial glycemic and venous serum insulin responses. Invertase, glucose oxidase, and catalase were used to catalyze the conversion of glucose in apple juice to gluconate/D-gluconolactone. The absorption of glucose in the upper small intestine of rats was completed, while the absorption of gluconate was 20%. Glycemic load (GL), glycemic response (GR), and sugar content of juice after enzymatic treatment decreased by 74.6%, 68%, and 21%, respectively. This phenomenon is due to the reduction of glucose content. For sensory evaluation, there was no significant difference in the sweetness of juices after enzymatic treatment compared with the untreated juices, although gluconic acid has a bitter taste. This is due to the difference in the sweetness level of the sugars. If sucrose sweetness is set at 100, fructose sweetness is 173, and glucose sweetness is 74 [91].

#### 2.3.2. Prebiotic Oligosaccharides

Prebiotics belong to functional food ingredients that stimulate a healthy microbiota. The major class of prebiotics is non-digestible oligosaccharides (NDOs), generally not digested in the upper intestine and the stomach but operating at the colon level. NDOs can be obtained during enzymatic synthesis. The enzymes are most often used to produce NDOs. These include glucosyltransferases that catalyze the transfer of carbohydrate residues between the activated donor and acceptor. These enzymes can be subclassified as retaining or inverting, depending on maintaining the stereochemistry of the glycosidic bond (α/β). The origin of the enzyme determines the type of glycosidic linkages and the type of reaction product [70].

Among prebiotic oligosaccharides, which can be obtained directly in the juices, fructo-oligosaccharides (FOS) and gluco-oligosaccharides (GOS) with prebiotic potential can be distinguished. Gluco-oligosaccharides are composed of β-D-glucose subunits linked by α-(1–6) and α-(1–2) bonds (indigestible to human gastric enzymes) [69]. Fructo-oligosaccharides are formed from fructose units linked by β-(2–1) glycosidic bonds and a terminal glucose unit. If the structure of FOS contains only a few fructose units, it is named a short-chain fructo-oligosaccharide (scFOS). The group of scFOS consisted of 1-ketose (trisaccharide), nystose (tetrasaccharide), and fructosylnystose (pentasaccharide) (Figure 3) [70]. Fructo-oligosaccharides improve nutritional and functional properties as essential ingredients of functional food [92]. Oligosaccharides have a positive impact on extending fresh fruit shelf life and preserving postharvest quality [93].

Among the enzymes capable of catalyzing reactions that lead to obtaining prebiotic oligosaccharides, fructosyltransferases and glucosyltransferases can be distinguished (Table 4). Fructosyltransferases (FTF) re glycoside hydrolases produced by fungi, bacteria, and plants. FTF is responsible for the synthesis of fructo-oligosaccharides from sucrose [94]. Glucosyltransferases (GTase; sucrose 6-glucosyltransferase) synthesize glucans from sucrose [95]. GTase can be produced by *Streptococcus oralis* (an early-colonizing microorganism in the oral cavity of humans)*, Streptococcus mutans*, or *Streptococcus sobrinus* (the major causative agent of dental caries) [95,96]. Dextransucrase, belonging to the family of glucosyltransferases, polymerizes the glucose molecules in sucrose into soluble and insoluble poly glucans and releases fructose [97] by glucosyl transfer [98].

According to a patent, WO/2012/059554 is a method of enzymatic conversion of sugars in juices and ready-to-drink (RTD) drinks into non-digestible carbohydrates and oligosaccharides such as gluco-oligosaccharides and fructo-oligosaccharides. Reducing the sugar content in juices can be obtained using glucosyltransferase and/or fructosyltransferase. Glucosyltransferases (such as a dextransucrase) catalyze gluco-oligosaccharides—non-digestible carbohydrates. A dextransucrase treatment of sucrose usually produces dextran and fructose as a side product. Oligosaccharides can be synthesized while other carbohydrates (acceptors) are present in addition to sucrose. Fructosyltransferase enables the conversion of sucrose into fructo-oligosaccharides (FOS). FOS has Generally Recognized As Safe (GRAS) status. FOS has documented health benefits and belongs to a group of prebiotics. Glucosyltransferases and fructosyltransferases should contact the juice sequentially. The preferred embodiment is first to convert sucrose into FOS and, in the next stage, convert sucrose and glucose into GOS. Controlled use of glucosyltransferase (such as dextransucrase) and fructosyltransferase enables a nutritious product with higher fiber content, thus enabling the classification of the juice as “sugar-reduced”. The total sucrose content reduction after exposure to fructosyltransferase and dextransucrase compared to a corresponding juice is at least 30% [101].

Patent US 2018/0146699 A1 describes reducing a concentration of disaccharides and/or monosaccharides in food material (in fruit juices containing monosaccharides and disaccharides, such as apple or orange juices) using glucosyltransferase. The process included the addition of Ca^2+^ ions to stimulate the enzyme’s activity. The reaction should be carried out at a pH of 3–5 and preferably a temperature of 45–55 °C (the optimal temperature is 50 °C). The glucosyltransferase should be immobilized through support before contact with a food product, using one of the known techniques: covalent binding, entrapment, physical adsorption, and cross-linking. Applying heat, conducting pasteurization, or removing the immobilized enzyme from contact with the food product may be used to terminate the glucosyltransferase enzymatic reaction. During the process, monosaccharides and/or disaccharides are converted into oligosaccharides (gluco-oligosaccharides) and/or polysaccharides. After the process, the saccharose content may be reduced by 10–99% [100].

Nguyen et al. (2015) used dextransucrase from *L. mesenteroides* with calcium hydroxide (to improve enzyme activity) to convert sugars in concentrated orange juice into oligosaccharides. The authors tested different concentrations of the enzyme (0.9–4/4 U/mL) and Ca(OH)_2_ (0.1–2% *w*/*v*). The optimum conditions of 3.52% U/mL dextransucrase and 1% Ca(OH)_2_ were selected. In this study, over 97.8% of the sucrose was enzymatically converted into GOS, and the synthesized yield of oligosaccharides was over 30% [26].

Tingirikari et al. (2017) applied the immobilization of dextransucrase from *L. mesenteroides* and co-immobilization with dextranase from *C. erraticum* using alginate beads and the entrapment technique. The optimum alginate concentration for the immobilization of the enzyme was 2.5% (*w*/*v*) with a 96% immobilization yield and 2.4 IU/mg of enzyme activity. The authors indicated that co-immobilization has a beneficial effect on process efficiency. Immobilized dextransucrase produced 37 g/L of GOS, and co-immobilized dextransucrase produced 41 g/L of GOS. Additionally, orange juice with pulp obtained 10% more oligosaccharides than orange juice without pulp [99].

Hajar-Azhari et al. (2020) used the commercial enzyme Viscozyme^®^ L from Novozymes as a biocatalyst for FOS synthesis in sugarcane syrup. Viscozyme^®^ L is a blend of pectinases, hemicellulases, xylanases, and beta-glucanases [104]. The enzyme concentration was 4% (*v*/*v*) (36 FU/mL; FU-fructosyltransferase units). Enzymatic synthesis was conducted in a specialized pH-stat bioreactor (1 L—SPSB system) at a pH of 5.5, T = 50 °C. After a 6 h reaction, the enzymatic yield was 32.2% of FOS (1-ketose) [102].

The study by Ureta et al. (2019) presents a synthesis of fructo-oligosaccharides in grapes using Viscozyme^®^ L from Novozymes. Grapes contain only glucose and fructose; therefore, different concentrations of sucrose (20% (*w*/*w*), 30% (*w*/*w*), and 55% (*w*/*w*)) were added to the oxidized grapes. In addition, 20% (*w*/*w*) of sucrose was insufficient to produce the synthesis of fructo-oligosaccharides since only 4% of the composition of products represented FOS (1-ketose, nystose). In the case of higher concentrations of sucrose, similar amounts of FOS were produced (21.8% and 21.0% for 30% and 55% of sucrose, respectively) [105]. The results obtained in that study helped choose juices of a sufficiently high concentration of sucrose to synthesize FOS.

Generally, prebiotics positively affect sensory properties such as taste or smell [106]. A sensory evaluation of pineapple, orange, and mango fruit juices fortified in fructo-oligosaccharides showed no difference in color, taste/flavor, and overall quality compared with juices with the addition of sucrose. The sweet taste of sucrose and FOS is very similar. Moreover, juices enriched in FOS were successfully stored at ambient or refrigeration temperatures for 4–6 months without decreasing the overall physicochemical and sensory quality [107].

#### 2.3.3. Enzyme Treatments and Metal Ion Supplementation

Sensory analysis of fruit juices in which free glucose was removed or partially reduced and converted into gluconic acid showed that the taste of the obtained product was lower than expected, and the incorporation of flavors or sweeteners appeared necessary. This juice with low sugar content does not leave a mouthfeel of the untreated juice. If, in addition, fructose and/or sucrose were removed, the juice’s sensory score was even lower. Appropriate metal ions were added to the reduced sugar juice to improve the sensory quality of juice with low sugar and at least 5 g/L gluconic acid. Results showed that the treated juice exhibited excellent sensory properties; the treated juice’s flavor, taste, and mouthfeel are comparable to that of untreated juice [108]. The European Union subjects of Regulation (EC) No 1925/2006 of the European Parliament establishes a list of vitamins and minerals (and their chemical forms) that may be added to food products. The addition of minerals in their bio-available form is permitted even if they are usually contained in the juices [109].

The patent WO/2016/051190 [108] describes obtaining juices with low sugar content and adding selected ions. This sugar-reduced juice was obtained as a result of enzymatic reactions. In the first stage, saccharose was hydrolyzed to fructose and glucose. Then, the glucose in the juices was converted into gluconic acid. In the next stage, at least two metal ions may be added to the juice: Ca^2+^, K^+^, or/and Mg^2+^. As a result of the described procedures, at least 5 g/L gluconic acid was formed. The patent provides detailed guidelines for various juices. In the case of apple juice, these requirements are given below:(a)At least 1.5 g/L K^+^,(b)At least 0.5 g/L Ca^2+^,(c)At least 0.1 g/L Mg^2+^.

The exact total sugar content of the final sugar-depleted juice product depends on the raw material. It can be reduced by 0–70%, usually 10–50%, and most preferably by 25–35%. The concentration of glucose and, optionally, sucrose content was reduced to trace levels, but the content of fructose was increased due to the conversion of sucrose into glucose and fructose. A juice product prepared by using this method may also increase insulin sensitivity. The juice with reduced sugar content and supplemented by metal ions (such as Ca^2+^, K^+^, and Mg^2+^) are suitable for treating and preventing complex metabolic disorders associated with the over-consumption of glucose and/or sucrose or the inappropriate metabolism of glucose, including diabetes, obesity, and insulin resistance. The juice has therapeutic properties and superior nutritional value [108].

#### 2.3.4. Low-Calorie Compounds

D-allulose (D-psicose, D-ribo-2-hexulose) is an epimer of D-fructose at the C-3 position. This rare sugar naturally occurs in small quantities, such as in kiwi fruits, raisins, or figs. D-allulose is a natural, almost calorie-free sugar (0.4 kcal/g) and has 70% of the relative sweetness of sucrose [66]. Therefore, D-allulose can be used as a sweetener in controlling a body weight diet. It is recommended for people with diabetes since it does not affect insulin [110] and does not significantly attenuate the increase of glucose levels in the blood [111] (which allows for glycemic control). Moreover, it is tooth-friendly, in contrast to sugar. The taste of D-allulose is similar to sugar, contrary to many other sugar substitutes, so consumers perceive it positively [110]. In the US, allulose has been approved as safe by the Food and Drug Administration since 2012 (GRN No. 400). It can be used as a sugar substitute in a variety of food categories, for example, non-carbonated beverages (max concentration—3.5%) [112]. It is still under consideration in Europe before the European Food Safety Authority (EFSA) approval because it has been classified as previously unavailable in food [110]. However, despite the existence of beneficial aspects of the use of allulose, in vitro studies have shown that bacteria such as *Klebsiella pneumonia* (an opportunistic human pathogen) can utilize allulose as a substrate. This raises the question of whether a high dietary intake of D-psicose may contribute to the growth of potentially harmful bacteria at mucosal sites such as the intestine [113].

The biosynthesis of D-allulose is carried out using the Izumoring strategy. D-fructose can be converted into D-allulose with the use of ketose 3-epimerases (KEases), which can be classified as D-allulose-3-epimerase (DAE) and D-tagatose 3-epimerase (DTE) [103].

There is only one publication concerning transforming D-fructose in fruit juices into D-allulose. C. Li et al. (2021) focused on orange juice, mango juice, and sugar cane juice to obtain D-allulose from sugars presented in these juices by using free invertase combined with immobilized D-glucose isomerase and immobilized thermostable D-allulose-3-epimerase from *Pirellula* sp. (PsDAE) in a two-step synthesis (Figure 4). This pathway enables obtaining D-psicose from sucrose, acting as a substrate. Invertase catalyzes the hydrolysis of sucrose into glucose and fructose. D-glucose isomerase catalyzes the isomerization of D-glucose into D-fructose. In the last stage, PsDAE was immobilized onto the epoxy support, which improved its thermal stability, and D-fructose was converted into D-allulose. D-allulose-3-epimerase from *Pirellula* sp. is dependent on the presence of a metal cofactor. Activity of PsDAE increased by using Co^2+^, Mg^2+^, Mn^2+^ ions by 1.75-, 1.28-, and 1.17-fold, respectively. Nevertheless, in that study, no element was added during the reactions. Finally, C. Li et al. obtained 16.4–19.3% of D-allulose among the total monosaccharides presented in these juices. The reaction of obtaining D-allulose is a very difficult and time-consuming process without industrial application [103]. The main restriction of industrial production of D-allulose is that at most 30% of D-fructose is converted into D-psicose. In other words, the reaction equilibrium between D-allulose and D-fructose is 30:70 [114].

According to the Izumoring strategy, D-psicose is the only rare sugar that can be produced directly from D-fructose [35]. Lee et al. (2017) proposed a method for the biotechnological production of tagatose from fructose, a C-4 epimer form of D-fructose [115]. The sweetness of tagatose is 92% of sucrose sweetness. D-tagatose provides a caloric value of 1.5 kcal/g. In the USA, D-Tagatose is generally considered safe (GRAS) as a sweetener for use in food [116]. The authors used a three-step enzymatic cascade reaction: hexokinase, fructose-1, 6-biphosphate aldolase (FbaA), and phytase. Conversion of fructose to tagatose was 80% [115]. This strategy could have been useful in fruit juices. Drinks containing D-psicose have improved taste qualities [117].

The last possible pathway for reducing the sugar content in juices is converting fructose into sorbitol. This compound provides a caloric value of 2.6 kcal/g (30% lower in comparison with sucrose) and has sweetness 40% lower than sucrose. Sorbitol belongs to a group of sugar alcohols and is used in the food industry as a nutritive sweetener. Like other sugar alcohols, sorbitol can negatively affect the digestive system, causing gastrointestinal distress [67,118]. The study conducted by Aziz et al. (2011) describes an approach for producing low-sugar pineapple juice. The authors used invertase (3.5 U/mL) to convert sucrose into glucose and fructose. In the last stage, glucose-fructose oxidoreductase (GFOR) from *Zymomonas mobilis* was used to oxidize glucose to gluconolactone, rapidly hydrolyzing to gluconic acid (4% *w*/*v*) and simultaneously converting fructose into sorbitol. After a 24 h reaction at pH 6.2 (optimum for GFOR), the conversion of sugars was about 30%. Since reaction with GFOR should be carried out at a relatively high pH, this process might be more appropriate for juices with naturally higher pH, such as topical fruits (mango, jackfruit, papaya) [68]. The sensory evaluation of starfruit juice with reduced sugar content with the addition of different ratios of cane sugar and sorbitol (from 0% to 10% for each sweetener) showed that a high concentration of sorbitol had a negative impact on the color of the juice (it was too pale). This is explained by the fact that sorbitol does not cause Maillard reactions like glucose or fructose do [119]. Moreover, higher sorbitol levels with simultaneously lower sugar cane levels caused lower taste preference of the juice because it was too sour [120].

## 3. Current Challenges and Future Perspectives

Juice producers use most types of mentioned processes. Membrane processes (microfiltration and ultrafiltration) are widely used in juice processing to clarify juices [121], whereas nanofiltration is used in the concentration and separation of bioactive compounds from juices (such as polyphenol concentration) [122,123]. In the food industry, enzymes are used very widely. These include pectinases, cellulases, xylanases, α-amylases, esterases (for enhancement of fragrance and flavour in fruit juices), and debittering enzymes [124]. However, due to the fact that producing juices with reduced sugar content is not allowed in the EU, these processes are not directly used to reduce the sugar content in fruit juices.

The partial removal of sugars from fruit juices without deteriorating their biofunctional properties and without the formation of by-products is a very difficult technological challenge. Different from small-scale production, mass production of such products poses additional challenges. Separation processes in their classical form may not be sufficient for this due to the phenomena of fouling and retaining bioactive compounds on the surface of the membrane, which is a bottleneck of these technologies. Limitations of conventional separation processes contribute to increasing production costs, which deteriorates the accessibility of juices to the general consumer following the membrane process. Improving regeneration and the existing membrane cleaning methods are essential for the application of membrane filtration processes in an industrial setting. Moreover, in the case of membrane processes, achieving a minimum loss of valuable bioactive substances in juices is challenging during the sugar-lowering process. Developing these technologies, including functional membranes, can improve the separation of sugars in complex feed systems, including in juices, which can be improved by modifying operation parameters such as feed flow or pressure.

While juice’s natural composition can change due to disrupting compounds as a result of enzymatic methods, additionally, reusing enzymes is problematic. It is worth exploring the possibilities of immobilization enzymes instead of free enzymes. This is attributed to the possibility of multiple uses in the production process, which generates lower operating costs. Reducing the adverse effect of immobilization, such as loss of enzyme activity connected with limited availability in the enzyme’s active site, is a challenge. To achieve improvement in this respect, it is necessary to examine the possibility of improving the efficiency of enzymatic reactions by investigating the synergic effect of several technologies, including emerging methods, such as the application of a pulsed electric field (PEF). A PEF can be successfully used to improve the activity of certain types of enzymes. Still, selecting the appropriate process conditions is very important because excess PEF may also lead to the deterioration of enzyme activity. Moreover, further exploring new potential enzymatic methods and conversion paths of saccharides is also important. There are few publications focused on reducing the sugar concentration directly in juices. This issue is particularly relevant to fructose and glucose because they are still poorly understood.

Additionally, maintaining the palatability of juices after a sugar reduction process is a serious challenge. Good quality and a proper balance of flavors make people willing to buy juices. As expected, a deterioration of sensory properties will result in low sales. Therefore, this matter requires special attention. In every case, consideration should be given to determine the most favorable degree of sugar reduction. Additionally, juice’s sensory quality after various sugar transformation processes should be investigated. It will allow for the determination of optimal transformation methods. Moreover, as mentioned before, adding selected ions can improve the sensory quality of juice. Therefore, the impact of all potentially useful ions should be examined.

## 4. Conclusive Remarks

Excessive sugar consumption is associated with an increased risk of diet-dependent diseases. This phenomenon is the cause of looking for ways to reduce sugar levels in food products such as juices. The main challenge in producing sugar-reduced juices is maintaining an initial fruit juice composition to the greatest extent possible. As can be seen, all described methods require refinement to obtain high-quality products in industrial conditions. Looking for technological innovations that permit obtaining juices with unchanged composition, excluding sugar content, is inevitable. This section is just entering the path of rapid development, so this particular issue is still open.

The use of different methods designed to reduce the sugar content of juices raises questions about whether the products of such transformations can still be called “juices”. It is because, with a decrease in the sugar concentration, the composition of juices also changes to a greater or lesser extent. Enzymatic treatment can lead to obtaining a juice with additional functional food ingredients, but it is simultaneously equal to a further change in an original juice composition. The potential advantages of such transformations (such as improving functional and nutritional properties) cannot be an excuse for this problem. It can raise questions connected with the legitimacy of these conversions. This matter requires a very careful and detailed rethink. This issue will be clear only when the EU guidelines of permitted changes appear.

## Figures and Tables

**Figure 1 foods-12-01181-f001:**
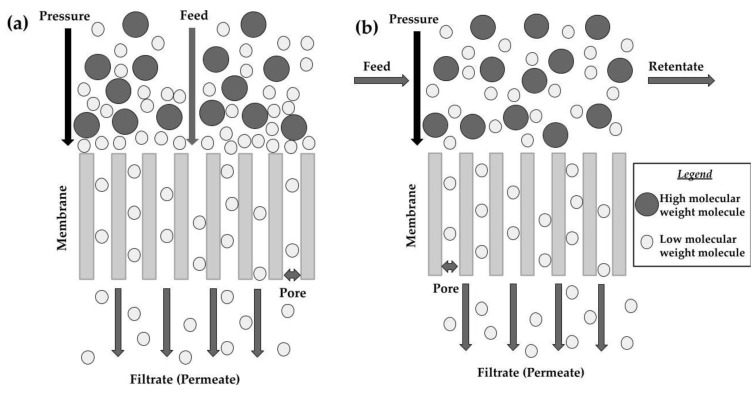
(**a**) Dead-end and (**b**) cross-flow filtration.

**Figure 2 foods-12-01181-f002:**
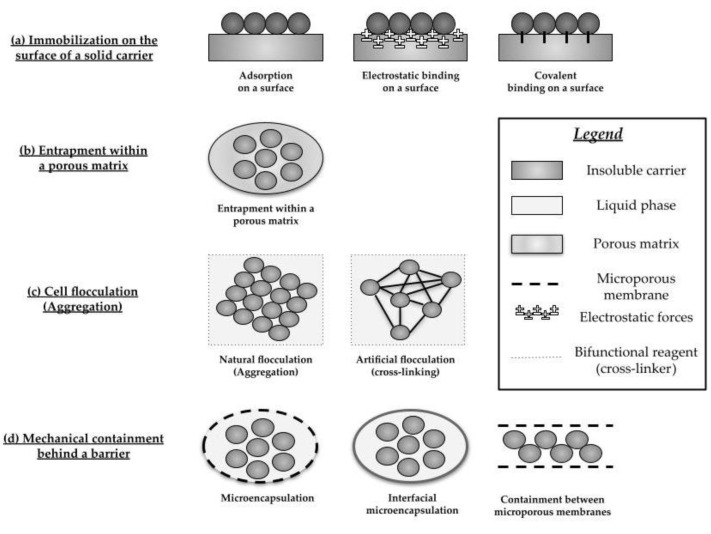
Basic methods of enzymes/cells immobilization.

**Figure 3 foods-12-01181-f003:**
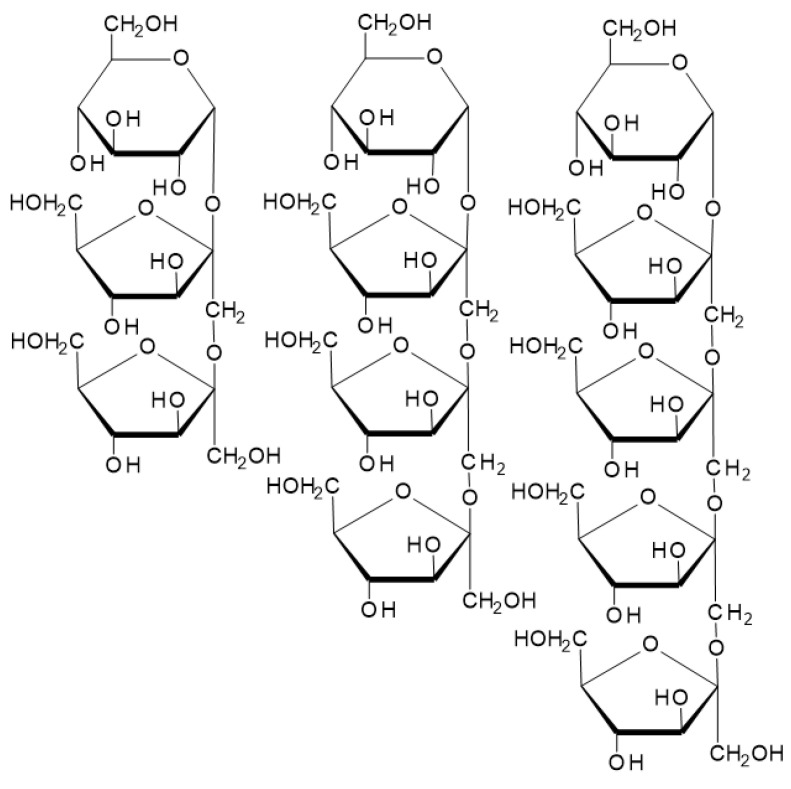
Chemical structure of scFOS produced from sucrose: 1-ketose, nystose, and 1F-fructofuranosyl nystose.

**Figure 4 foods-12-01181-f004:**
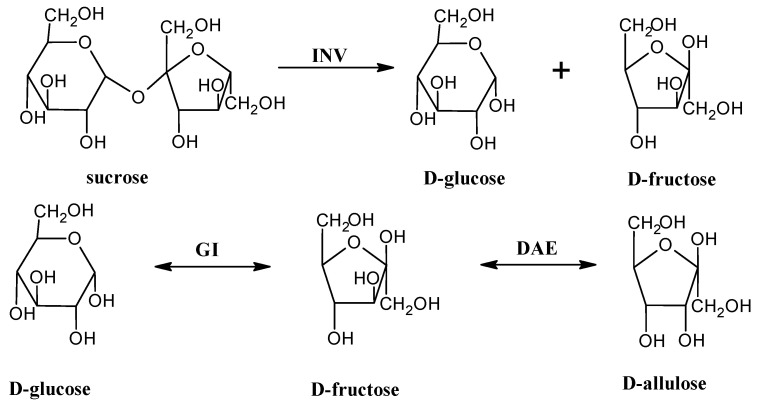
Schematic diagram of the multi-enzyme cascade reaction using INV (invertase), GI (glucose isomerase), and DAE (D-allulose-3-epimerase).

**Table 1 foods-12-01181-t001:** Comparison of sugar reduction levels after mixing two different juices.

Juice 1 (°Brix Value)	Juice 2 (°Brix Value)	Volume Ratio Juice 2/Juice 1 (*v*/*v*)	The Rate of Juice 2 °Brix Value Depletion
Coconut water (°Brix value equal to 3.6)	Apple/orange juice (°Brix = 10.0)	1:1	32%
		1:2	43%
	Grape juice (°Brix = 13.5)	1:1	37%
		1:2	49%
	Mango juice/puree (°Brix = 14.0)	1:1	37%
		1:2	49%
	Banana juice/puree (°Brix = 20.0)	1:1	49%
		1:2	55%
Acerola juice/puree or Cranberry juice (°Brix value equal to 6.0)	Apple/orange juice (°Brix = 10.0)	1:1	20%
		1:2	27%
	Grape juice (°Brix = 13.5)	1:1	28%
		1:2	37%
	Mango juice/puree (°Brix = 14.0)	1:1	29%
		1:2	38%
	Banana juice/puree (°Brix = 20.0)	1:1	35%
		1:2	47%
Strawberry juice/puree or Raspberry juice (°Brix value equal to 7.0)	Apple/orange juice (°Brix = 10.0)	1:1	15%
		1:2	20%
	Grape juice (°Brix = 13.5)	1:1	24%
		1:2	32%
	Mango juice/puree (°Brix = 14.0)	1:1	25%
		1:2	33%
	Banana juice/puree (°Brix = 20.0)	1:1	33%
		1:2	43%

**Table 2 foods-12-01181-t002:** Comparison of membranes possibility of a reduction of sugar content in fruit juices.

Process	Membrane and Pore Size	Type of Fruit Juice	Operating Conditions	A Degree of TS and TSS Reduction	Ref.
MF	Polyacrylonitrile-based MF-grade hollow fibers. Pore size: 0.1 μm	Sugarcane juice	T = 20 °C CFR = 30 L/h TMP = 104 kPa	TSS = 9.0%	[41]
MF	Mixed cellulose esters membrane (MCE), 0.45 µm	Pomegranate juice	-	TSS = 8.1%	[47]
MF	Hollow fiber membrane, 0.45 µm	Jamun (*Syzygium cumini*) juice	T = 30 °C CFR = 10 L/h TMP = 137.8 kPa	In retentate: TSS = 8.1%	[44]
MF	Mixed cellulose esters (MCE) membrane, 0.22 µm	Pomegranate juice	-	TSS = 24.6%	[45]
UF	Mixed cellulose esters (MCE) membrane, 0.025 µm	Pomegranate juice	-	TSS = 20.4%	[45]
MF/UF	Mixed cellulose esters (MCE) MF membrane, 0.22 µm Mixed cellulose esters (MCE) UF membrane, 0.025 µm	Pomegranate juice	-	TSS = 40.4%	[42]
UF	Ceramic membrane, 15 kDa	Cloudy apple juice	TMP = 0.35 MPa	In permeate: TS (after UF) = 21.2% TSS (after UF) = 20.0% In retentate: TS (after UF) = 15.4% TSS (after UF) = 16.4%	[48]
UF	Three hollow fiber membranes: Polysulfone (PS), 100 kDa Polysulfone (PS), 50 kDa Polyacrylonitrile (PAN), 50 kDa	Red-colored blood oranges (*Citrus sinensis*) juice	T = 20 °C TMP = 50 kPa Q_f_ = 140 L/h	TSS = 2.0% TSS = 2.0% TSS = 1.9%	[49]
UF	Polyethersulfone, 10 kDa (PES-10 kDa)	Apple juice	T = 25 °C CFR = 30 L/h TMP = 0.75 MPa	TS = 54.3% TSS = 31.7%	[50]
UF	Polysulfone, 100 Da	The Xoconostle fruits	T = 25 °C TMP = 138 kPa Q_f_ = 58 L/h	TSS = 10.2%	[51]
Primary clarification MF + UF	Hollow fiber polyacrylonitrile (PAN) MF membrane Polymeric hollow fiber membrane made of polysulfone, 30 kDa (PSU-30 kDa) UF membrane	Kinnow (mandarin) juice	T = 25 °C TMP = 69 kPa CFR = 20 L/h	TS = 1.4% TSS = 13.4%	[52]
UF/DF	Ceramic tubular membrane, 15 kDa	Cloudy apple-cranberry juice	TMP = 0.35 MPa	In permeate: TS (after UF) = 25.5% TSS (after UF) = 22.6% TS (after DF) = 31.6% TSS (after DF) = 45.2% In retentate: TS (after UF) = 23.5% TSS (after UF) = 16.1% TS (after DF) = 41.8% TSS (after DF) = 34.7%	[53]
NF/DF	A flat sheet polyamide-thin film composite NF membrane, 150–300 Da	Apple juice	T = 25 °C Pressure = 50 bar flow rate = 40 L/h	TS = 94.9%	[19]
NF	Spiral polyvinylidene difluoride (PVDF) membrane, 150–300 Da	Watermelon juice	T = 25 °C pressure = 600 kPa flow rate = 1 m/s	TSS = 29.4%	[54]
Clarified by UF + NF	Microdyn Nadir Polyethersulfone, 1000 Da	Bergamot juice	TMP = 6 bar T = 20 °C	TSS = 33.6%	[55]
	Microdyn Nadir Polyethersulfone, 400 Da			TSS = 52.7%	
	Semi-aromatic piperazine-based polyamide layer on top of a polysulphone microporous support, 150–250 Da			TSS = 75.3%	
Clarified by UF + NF/DF	Spiral-wound membranes: TFC 200–300 Da	Apple juice	T = 25 °C TMP = 25 bar Q_f_ = 7 L/min	TS = 60%	[56]
MF/UF/NF	Hollow fibre MF/UF membranes, 0.45 µm (MF) and 50 kDa (UF). Spiral wound NF membrane, 300 Da	Indian blackberry juice	MF/UF: TMP = 0.137 mPa NF: TMP = 2.5 mPa	TSS = 16.7%	[57]

Legend: MF, microfiltration; UF, ultrafiltration; NF, nanofiltration; DF, diafiltration process; TMP, transmembrane pressure; CFR, cross flow rate; Q_f_, axial feed flow rate; FR, flow rate; TS, total sugar, TSS, total soluble solids; T, temperature.

**Table 3 foods-12-01181-t003:** Comparison of immobilization methods.

Basic Methods of Enzyme/ Cell Immobilization	Characteristics of the Method	Examples of Carriers
Immobilization on the surface of a solid carrier	Consists of the creation of a covalent bond between the cell membrane of the microorganism/the enzyme and the carrier or is the result of electrostatic forces on the carrier, causing physical adsorption	Cellulosic materials: DEAE-cellulose, wood, delignified sawdust, sawdust; inorganic materials: porous porcelain, hydromica, porous glass, palygorskite, montmorillonites
Entrapment within a porous matrix	The porous material is formed into the cell culture, and the cells or the enzyme are allowed to penetrate the porous matrix until other cells/enzymes restrict their mobility	polysaccharide gels: chitosan, polygalacturonic acid, alginates, κ-carrageenan, agar; polymeric matrixes: polyvinyl alcohol, collagen, gelatin
Cell/Enzymes flocculation (Aggregation)	It is a cell/enzyme aggregation of the microorganism/enzyme: physical or chemical cross-linking. It is an aggregation of cells/enzymes to form a larger unit. Because of the large aggregates, they can be used as a method of immobilization	Mainly molds, fungi, and plant cells are capable of forming aggregates
Mechanical containment behind a barrier	Using microporous membrane filters, we obtained immobilization of a cell/enzyme into a microcapsule or entrapment of a cell/enzyme on the interaction surface of two liquids that are not miscible. An appropriate type of immobilization is when little transfer of compounds and cell-free products is expected. The main disadvantage of this method is that the growth of cells can fill the filter and the limitation of mass transfer	Chitosan, k-carrageenan, and collagen can be used as polymers’ porous networks for entrapment

**Table 4 foods-12-01181-t004:** Comparison of enzymes applied to reduce sugar content in juices.

Sugar	Enzyme	Product	Type of Juice	Additional Compounds	Ref.
Sucrose Glucose	Invertase; Glucose oxidase; catalase	Gluconic Acid	Apple juice	KOH/Ca(OH)_2_ to optimize organoleptic properties	[91]
Sucrose Glucose	Dextransucrase from *L. mesenteroides* and dextranase from *C. erraticum*	GOS	Orange juice	Co-immobilization of enzymes on alginate beads	[99]
Sucrose Glucose	Dextransucrase from *L. mesenteroides*	GOS	Concentrated orange juice	1% Ca(OH)_2_ to improve transferase activity, hydrolytic activity, and the total activity of the enzyme	[26]
Sucrose Glucose	Glucosyltransferase that comprises an amino acid sequence at least 95% identical to SEQ ID NO: 1	GOS	Fruit juice containing sucrose, and glucose/fructose	1 mM CaCl_2_ to improve transferase activity, hydrolytic activity, and the total activity of the enzyme	[100]
Sucrose Glucose	Glucosyltransferase (such as dextransucrase)	GOS	Fruit juice containing sucrose and glucose	-	[101]
Sucrose	Fructosyltransferase	FOS	Fruit juice containing sucrose		
Sucrose	Viscozyme L (Novozymes, Denmark)	FOS	Sugarcane syrup	-	[102]
Sucrose Glucose Fructose	Invertase (INV) GFOR from *Z. mobilis*	Gluconic acid; Sorbitol	Pineapple juice	-	[68]
Sucrose Glucose Fructose	Invertase (INV); D-glucose isomerase (GI); D-allulose 3-epimerase from *Pirellula* sp. (DAE)	D-Allulose	Mango, orange, and sugar cane juices	GI and DAE were immobilized on epoxy support	[103]

Legend: GOS, gluco-oligosaccharides; FOS, fructo-oligosaccharides; GFOR, glucose-fructose oxidoreductase; -, no information available.

## Data Availability

Data is contained within the article.

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
