# Peer review of "Application of Emerging Techniques in Reduction of the Sugar Content of Fruit Juice: Current Challenges and Future Perspectives"

_foods, 2023, doi:10.3390/foods12061181_

Round 1

Reviewer 1 Report

This review is interesting. Some improvements should be considered by authors to increase the quality and clarity of the information.

Pg. 4-7: The parts with examples of specific juice fruits, from literature, should be presented in table format to improve the possibility of comparison of different membrane processes and its main results.

Pg. 13 – The authors should explain why only D-allulose is mentioned across the sweeteners available in the food market.

In discussion, the authors should clarify if all the methods present in manuscript are already used by juice producers or explain which methods are not allowed. Moreover, they should explain which methods are currently more frequent in fruit juices industry.

Author Response

Dear Reviewer,

Thank you very much for your comments on our paper. All comments and remarks were valuable and helpful in improving our paper. We have studied the comments carefully and have made corrections which we hope meet with approval. 

Reviewer 2 Report

Comments to the author;

There is a challenge in producing sugar-reduced juices while preserving the initial fruit juice composition and its nutritional compounds. In this article, Emerging methods for producing reduced sugar juices are reviewed, including membrane and enzymatic processes. Technological innovations are needed in this area as the existing methods have limitations and need improvement.

The methods discussed include both membrane and enzymatic processes. The limitations and challenges of these methods are also addressed, including the losses of nutritional ingredients in membrane processes and the emergence of side products in enzymatic processes. The review also provides an overview of the current state of the field and identifies areas that require further research and technological innovations.

One concern is that some important points are missing as highlighted in the attached report including the physiochemical/organoleptic modifications in the juices after sugar reduction. Also, stick to the main theme of the article as Methods for producing reduced sugar juices, limitations of existing methods, improving existing methods, and Selectivity of enzymatic processes. And remove all unnecessary information other than sugar reduction in juices. Only focus on the studies related to the title.

Further comments can be accessed in the attached file.

Author Response

Dear Reviewer,

On behalf of all the co-authors, thank you very much for the time you have spent on reviewing our paper. Please find in the attached file the answers for the pointed issues and information about the corrections made in the paper. I am very grateful for a thorough review of the work. All your comments and remarks were valuable and very helpful in improving the paper and we hope that it’s now suitable for publication.

Responses to all comments can be accessed in the attached file.

Reviewer 3 Report

Application of emerging techniques in reduction of the sugar content of fruit juice: current challenges and future perspectives

Comments:

14. Check the sentence.

19. re-using enzyme. Please also check this sentence.

23. sugar-reduced fruit juice. This keyword is not right, I think.

Rewrite the abstract in more synthesis manner as well with novelty.

32. Please check these papers, which is on reducing sugar, alternative of refined sugar with natural sugars:

Replacement of refined sugar by natural sweeteners: focus on potential health benefits. Heliyon. 8(9), e10711.

Stabilization and attributive amelioration of sugarcane juice by naturally derived preservatives using aonla and moringa extract. Food Science and Nutrition, 9(6), 3048-3058.

41. One refence is not enough for these types of statements, please check these papers:

                    Preparation and characterisation of novelty food preservatives by Maillard reaction between polylysine and reducing sugars

47-57. No reference for the Eu study. Proper citation should be added.

66. Need reference for this statement.

71. Reference is not according to the format.

81. Need reference for this WHO related statement.

Line no 106: Reference?

There is no need to discuss the mechanism of techniques, add more data about main theme of manuscript. Same with all techniques

Try to write discussion in connective manner, the whole discussion is very weak.

Line no 172: update reference. Only add latest studies.

Line no 181, 213: too old reference

          Natural Plant Extracts: An Update about Novel Spraying as an Alternative of Chemical Pesticides to Extend the Postharvest Shelf Life of Fruits and Vegetables

213. Reference is not according to the format.

221. Reference is not according to the format.

238-241. No reference. Small paragraph, please merge.

269. Reference is not according to the format.

287. Reference is not according to the format.

298-300. No reference. Small paragraph, please merge.

No reference in Table 1. It’s strange?

370. Reference is not according to the format.

Conclusion is too long, should be rewritten in more synthesize way.

Overall this paper is very well written, but the negative impact too much same information is given, very old references are present, moreover try to write the data in compact way. References are not according to the format. current challenges and future perspectives were not discussed in a separate heading.

Author Response

(The authors gave the same response as above.)

Round 2

Reviewer 3 Report

Very well revised. 

Good luck.